# DelRec: learning delays in recurrent spiking neural networks.

## Abstract

Spiking neural networks (SNNs) are a bio-inspired alternative to conventional real-valued deep learning models, with the potential for substantially higher energy efficiency. Interest in SNNs has recently exploded due to a major breakthrough: surrogate gradient learning (SGL), which allows training SNNs with backpropagation, strongly outperforming other approaches. In SNNs, each synapse is characterized not only by a weight but also by a transmission delay. While theoretical works have long suggested that trainable delays significantly enhance expressivity, practical methods for learning them have only recently emerged. Here, we introduce "DelRec", the first SGL-based method to train axonal or synaptic delays in recurrent spiking layers. DelRec leverages a differentiable interpolation technique to handle non-integer delays with well-defined gradients at training time. We show that trainable recurrent delays outperform feedforward ones, leading to new state-of-the-art (SOTA) on two challenging temporal datasets (Spiking Speech Command, an audio dataset, and Permuted Sequential MNIST, a vision one), and match the SOTA on the now saturated Spiking Heidelberg Digit dataset using only vanilla Leaky-Integrate-and-Fire neurons with stateless (instantaneous) synapses. Our results demonstrate that recurrent delays are critical for temporal processing in SNNs and can be effectively optimized with DelRec, paving the way for efficient deployment on neuromorphic hardware with programmable delays. Our code is available at `https://anonymous.4open.science/r/Recdel-4175`.

## 1 Introduction

Inspired by the architecture and dynamics of biological neural networks, Recurrent Spiking Neural Networks (RSNNs) provide a compelling and energy-efficient framework for processing time-varying data. Their recurrent structure enables them to maintain an internal state, integrating information over extended periods, an essential capability for tasks involving temporal dependencies, such as speech recognition and time-series prediction (Bellec et al., 2018). However, despite their promise, RSNNs remain underutilized in machine learning due to significant training challenges, particularly the pervasive issues of vanishing and exploding gradients.

Recent advancements have sought to address these limitations by enhancing the models of spiking neurons (Yin et al., 2021; Bittar & Garner, 2022; Baronig et al., 2025). Innovations such as adaptive leaky integrate-and-fire (AdLIF) models and other sophisticated neuron dynamics have led to notable improvements, achieving state-of-the-art performance on widely used spiking neural network benchmarks. Yet, these neuron-centric approaches put their emphasis on modulating current inputs, rather than reactivating past signals, which limits their ability to model extended temporal dynamics.

An alternative approach involves incorporating transmission delays in synaptic connections, a feature observed in biological systems. In the brain, these delays are modulated by myelin, an insulating sheet around axons that accelerates conduction speeds. Evidence further indicates that myelin levels are plastic (Monje, 2018). Delays enhance network expressivity: neurons detect coincident spike time latencies, which, in the presence of heterogeneous delays, correspond to arbitrary spike onset latency sequences. While previous studies have successfully demonstrated the benefits of using delays in feedforward connections (Shrestha & Orchard, 2018; Sun et al., 2022; 2023; Timcheck et al., 2023; Hammouamri et al., 2024; Deckers et al., 2024; Göltz et al., 2025; Ghosh et al., 2025), their

potential in recurrent connections remains largely unexplored. Recurrent delays could offer even greater advantages, facilitating self-sustained activity (see Fig. 1A), modeling long-term dependencies, and supporting complex patterns like oscillations and polychronization. Theoretical work by Izhikevich demonstrated that recurrent delays transform a neuron's differential equation, expanding the range of possible solutions and enabling richer dynamics. Additionally, recurrent delays may mitigate gradient challenges by implementing temporal skip connections, improving gradient propagation during training (see Fig 1B).

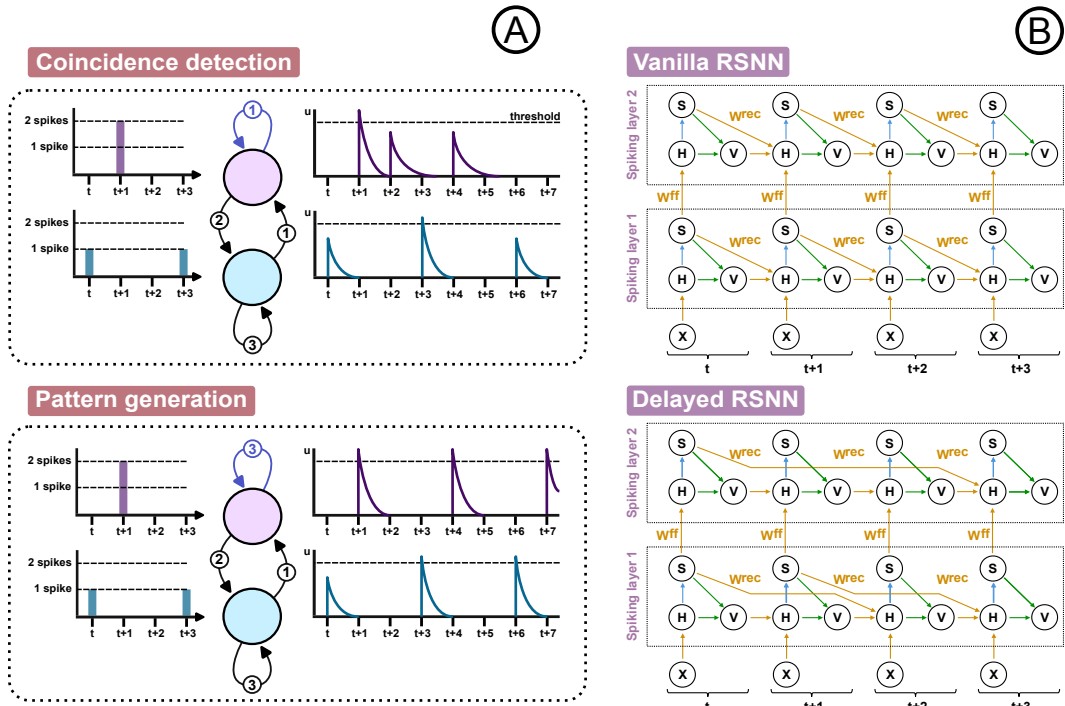

Figure 1: **A**: The optimization of a single delay in a recurrent connection can transform two recurrently connected neurons in a pattern generator. Two different behaviors of two neurons with the same inputs. Each neuron is recurrently connected to itself and to the other neuron, with a weight equal to 1. The recurrent connections each have a delay, indicated by the circled number on the connection. The neurons spike if they receive inputs strictly superior to 1 spike. At time $t - 1$, the neurons do not receive any input. The blue neuron receives an input spike at times $t$ and $t + 3$, while the pink neuron only receives two spikes at time $t + 1$. *Top*: The inputs trigger the firing of one spike per neuron, working as a coincidence detector for spikes reaching the two neurons in a short time interval. *Bottom*: When the delay of the pink neuron's recurrent connection (blue arrow) is increased from 1 to 3 time steps, the same input triggers a regular and sustained firing pattern. **B**: Delays in recurrent connections reduce the risks of exploding or vanishing gradients by bridging distant time steps. Computational graphs of a vanilla RSNN with a intrinsic delay of 1 time step in all recurrent connections (*Top*), and of a RSNN with different and longer delays in the recurrent connections (*Bottom*).

To date, however, only a handful of studies have explored the potential of delays in recurrent connections, and even fewer have focused on learning optimal delay configurations. Recent approaches have introduced algorithms to optimize these temporal parameters, demonstrating promising improvements in temporal tasks. For instance, Xu et al. achieved state-of-the-art results by learning a single recurrent delay parameter per layer using backpropagation. Their approach selects delays from a fixed set via a softmax function with a decreasing temperature, showcasing the potential of more flexible, parameterized methods. To the best of our knowledge, only Mészáros et al. (2025) have proposed an algorithm specifically designed to learn optimal delays in recurrent connections. Their method, tailored for the EventProp algorithm (Wunderlich & Pehle, 2021), leverages exact gradient computation. However, it inherits common limitations of EventProp, including scalability

challenges and suboptimal performance on real-world temporal benchmarks. Currently, all state-of-the-art spiking approaches on these benchmarks rely on surrogate gradient learning (SGL).

In this paper, we introduce "DelRec", the first method to train axonal or synaptic delays in recurrent connections using surrogate gradient learning (SGL) and backpropagation. Our method operates in discrete time and eliminates the need to predefine a maximum delay range. During training, we relax the constraint of integer delays by employing a differentiable interpolation process, then round delays to the nearest integer for inference. Our approach is implemented using the Pytorch-based Spikingjelly library (Fang et al., 2023), and is compatible with any spiking neuron model. We achieved new state-of-the-art results on two challenging temporal datasets (Spiking Speech Command, an audio dataset, and Permuted Sequential MNIST, a vision one), even when using simple Leaky-Integrate-and-Fire neurons. Additionally, our study is the first to combine the optimization of feedforward delays using DCLS (Hammouamri et al., 2024) and delays in recurrent connections. We believe this work establishes a foundation for leveraging delays in recurrent networks and provides an accessible tool to explore the potential improvements offered by such methods. Our results highlight the essential role of recurrent delays in temporal processing for spiking neural networks (SNNs). Using DelRec, these delays can be optimized effectively, opening new opportunities for deployment on neuromorphic hardware equipped with programmable delays.

## 2 METHODS

### 2.1 NEURON MODEL

Most spiking neuron models can be described by three discrete-time equations (Fang et al., 2021):

$$H[t] = f(V[t-1], I[t]), \tag{1}$$

$$S[t] = \Theta(H[t] - V_{\text{th}}), \tag{2}$$

$$V[t] = \begin{cases} H[t] \cdot (1 - S[t]) + V_{\text{reset}} \cdot S[t], & \text{if hard reset} \\ H[t] - V_{\text{th}} \cdot S[t], & \text{if soft reset} \end{cases}, \tag{3}$$

where $f$ is the neuronal charge function (which depends on the neuron model), $I[t]$ is the input current, $H[t]$ is the membrane potential after charging but before firing, $V[t]$ is the membrane potential after firing, and $S[t]$ is the output spike. $V_{\text{th}}$ in Eq. 2 is the threshold, and $V_{\text{reset}}$ in Eq. 3 is the reset potential. $\Theta(x)$ is the Heaviside step function, i.e. $\Theta(x) = 1$ for all $x \geq 0$, otherwise $\Theta(x) = 0$. We use the surrogate gradient method () defining $\Theta'(x) = \sigma'(x)$ during the backward step, where $\sigma$ is the surrogate function.

An example of a spiking neuron model, which is popular due to its simplicity, is the leaky integrate-and-fire neuron (LIF). The neuronal charge equation for the LIF neuron is

$$f(V[t-1], I[t]) = (1 - \frac{1}{\tau}) \cdot V[t-1] + \frac{1}{\tau} \cdot I[t], \tag{4}$$

where $\tau$ is the membrane time constant. We used the LIF neuron in all our experiments. Yet our method is compatible with any spiking neuron model that fits in the Eq. 1- 3 formalism.

Our method is essentially a way to compute $I$ in the presence of delayed recurrent connections. The total input current is the sum of the feedforward input $X$ and the recurrent one $X^{\text{rec}}$:

$$I[t] = X[t] + X^{\text{rec}}[t] \tag{5}$$

In a vanilla Recurrent Spiking Neural Network (RSNN), the outputs of each neuron of a layer $L$ at time $t$ are connected to the neurons of the same layer $L$ at time $t+1$. If we denote $\{S_j(t), j \in (L)\}$ the spikes emitted by all neurons of layer $L$ at time $t$, then we have for a neuron $i$ of the same layer:

$$X_i^{rec}[t] = \sum_{j \in (L)} w_{ij}^{\text{rec}} S_j[t-1] \tag{6}$$

where $\{w^{\text{rec}}\}$ are the local recurrent weights of layer $L$. We extend this regular definition of a RSNN by allowing neurons to project their outputs to neurons of the same layer, but with a longer delay. To this end, we set parameters $\{d_j \in \mathbb{N}, j \in L\}$ to model the time delay in a recurrent connection.

More specifically, if neuron $j$ in layer $L$ emits a spike $S_j(t)$ at time $t$, then neuron $i$ in layer $L$ will receive a recurrent input $w_{ij}^{\text{rec}} S_j(t)$ at time $t + 1 + d_j$. As a result, we modify Eq. 6:

$$X_i^{\text{rec}}[t] = \sum_{j \in (L)} w_{ij}^{\text{rec}} S_j[t - (1 + d_j)] \tag{7}$$

Here, for simplicity, we assume an identical delay for all outgoing connections of a given neuron. This is referred to as "axonal delay" in the literature, and we will use this setting in all our experiments. Yet our method/code is also compatible with synaptic delays (i.e., one different delay for each synapse). In the last equation, we used the arbitrary convention that a recurrent connection has a minimum delay of 1 time-step. Therefore, in our method, a delay parameter $d = 0$ leads to an effective delay of 1 in the recurrent connection.

## 2.2 LEARNING DELAYS IN RECURRENT CONNECTIONS

Let's consider one layer of $N$ different neurons, with input sequences of temporal dimension $T$. In order to learn the delay parameters $\{d_j \in \mathbb{N}, j = 1, ..., N\}$, we take a "future-oriented" perspective: when a neuron $j$ fires a spike at date $t$, we will schedule an input $w_{ij}^{\text{rec}}$ at date $t + 1 + d_j$ for all neurons $i$ of the same layer. To this end, we consider in practice $X^{rec} \in \mathbb{R}^{N \times T}$ as a scheduling matrix storing weighted spikes for future time steps. More specifically:

$$X_i^{\text{rec}}[t + \tau] = \sum_{j | 1 + d_j = \tau} w_{ij}^{\text{rec}} S_j[t] \tag{8}$$

In this framework, the parameters we learn in our method are the set of delays $d$ and the weights $w$.

However, for the purpose of optimization, we consider real-valued delays $\{d_j \in \mathbb{R}, j = 1, ..., N\}$, which leads to a modification of Eq. 8. If a spike is scheduled at date $t + 1 + d \in \mathbb{R}$, we temporally spread the prediction over time steps around $t + 1 + d$, with a triangle function $h_{\sigma,d}$ (Khalfaoui-Hassani et al., 2023) with a width parameter $\sigma$. More specifically, for all current time steps $t$ we consider the spread prediction at the target date $t + \tau$ :

$$h_{\sigma,d}(\tau) = \max(0, \frac{1 + \sigma - |\tau - (1 + d)|}{(1 + \sigma)^2}) \tag{9}$$

and we decrease the parameter $\sigma$ throughout training down to 0, as depicted in Figure 2C, so that by the end of training $h_{0^+,d}(\tau)$ leads to a linear interpolation between the two closest integer delay positions.

Using this method, at each time step $t$, we update future time steps of the scheduling matrix. So the prediction at a future time $t + \tau$ is updated (in the case of axonal delays) with:

$$X_i^{\text{rec}}[t + \tau] \leftarrow X_i^{\text{rec}}[t + \tau] + \sum_{j=1}^{N} w_{ij}^{\text{rec}} \cdot h_{\sigma_{\text{epoch}},d_j}(\tau) \cdot S_j[t] \tag{10}$$

$$\leftarrow X_i^{\text{rec}}[t + \tau] + \sum_{j=1}^{N} w_{ij}^{\text{rec}} \cdot \max(0, \frac{1 + \sigma - |\tau - (1 + d_j)|}{(1 + \sigma)^2}) \cdot S_j[t] \tag{11}$$

where $d_j$ is the axonal delay in the recurrent connection between neurons $j$ and $i$.

One can notice in Eq.15 that the function $h_{\sigma,d}(\tau)$ has a finite support $\text{supp}(h_{\sigma,d}(\tau))$, depending only of $\sigma$ and $d$. Indeed:

$$\forall \tau, \ h_{\sigma,d}(\tau) = 0 \Leftrightarrow \tau \in \text{supp}(h_{\sigma,d}) = \big[(1 + d) - (1 + \sigma) \ ; \ (1 + d) + (1 + \sigma)\big]$$
$$= \big[d - \sigma \ ; \ 2 + d + \sigma\big] \tag{12}$$

So, it is in fact sufficient to schedule recurrent inputs only in $\text{supp}(h_{\sigma,d})$, and as $\sigma$ decreases during training, the range of time steps when we can schedule recurrent inputs becomes narrower. At the scale of a layer, we need to schedule inputs for multiple neurons at the same time, which means it suffices to compute and schedule inputs for a limited range of time steps $\mathbf{E}$, such that:

$$\tau \in \mathbf{E}(\sigma, D) = \bigcup_{d \in D} \text{supp}(h_{\sigma,d})$$

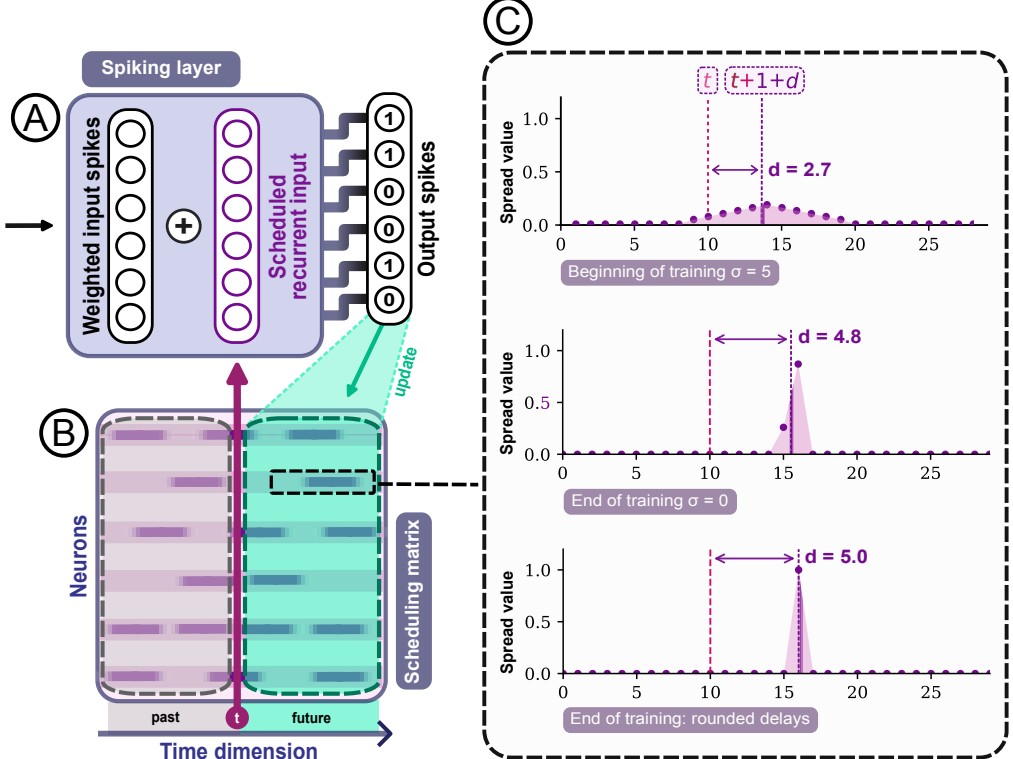

Figure 2: **A**: At time step $t$, the neurons in the studied layer receive weighted spikes from the previous layer, which are then summed with the inputs scheduled for time step $t$ in the scheduling matrix.. The subsequent evolution of the internal state of the neuron (membrane potential) may produce output spikes. **B**: Each neuron of the layer receives a weighted sum of the output spikes, and schedules it on a spread of future dates determined by its delay parameter $d$ and the spread of the epoch $\sigma$. Spread values are represented by the purple gradients. **C**: We modify the spread function at each epoch by reducing its $\sigma$. At the beginning of training, the scheduled values are widely spread around the true delay $d$. When the training ends, $\sigma$ is close to 0 and the spread function only performs linear interpolation between the closest integers from the floating delay. Then we manually round the delay to the closest integer in evaluation mode.

with $N$ the number of neurons in the layer, and $D = \{d_j \; ; \; j = 1, \; ... \; , N\}$. In practice, as $\sigma$ is decreasing to 0 in Eq. 12, we ignore the lower bound of $\mathbf{E}$, and we approximate this set with:

$$\tilde{\mathbf{E}}(\sigma, D) = \left[ \, 0 \, ; \; \left\lceil 1 + \max_{1 \leq j \leq N} d_j + (1 + \sigma) \right\rceil \, \right] \supset \mathbf{E}(\sigma, D). \tag{13}$$

In other words, we only need to compute and store $h_{\sigma,d}(\tau)$ for $\tau$ in $\tilde{\mathbf{E}}(\sigma, D)$, so the scheduling matrix $X^{rec}$ has in fact a dimension of $N \times \dim(\tilde{\mathbf{E}}(\sigma, D))$, then at time $t$, the future recurrent input at time $t + \tau$ is updated with Eq. 11. We use $X^{\text{rec}}$ as a buffer with a pointer mechanism in order to efficiently schedule the future recurrent inputs (see Algorithm 1).

The initial value of $\sigma$ allows the recurrent connections to capture broad temporal dependencies at the beginning of training and leads to a loose optimization of delays on long time scales, while the subsequent refining of $\sigma$ during training pushes the delay parameters towards more precise locations. A similar strategy was used in (Hammouamri et al., 2024). The described method is illustrated in Fig. 2 and leads to Algorithm 1, which describes the update of the internal parameters of our neuron.

## 3 RESULTS

### 3.1 STATE-OF-THE-ART PERFORMANCE ON AUDIO AND VISION TASKS

We evaluated our method on two state-of-the-art datasets: the SSC (Spiking Speech Commands) and the PS-MNIST (Permuted Sequential MNIST). The SSC dataset is a spiking audio dataset which demands leveraging temporal patterns in spike trains to reach good classification accuracies. While it is one of the most widely used datasets in the SNN community for benchmarking models' temporal processing capabilities, it also stands out as one of the largest, featuring over 100k samples across 35 classes of spoken commands. It is worth noting that this dataset has dedicated training, validation and test sets, and is far from saturated (with best accuracies around 80%). The PS-MNIST is a vision dataset which is obtained by flattening all images of the MNIST ($28 \times 28$) into one sequence ($1 \times 784$), and permuting the pixel positions. This transformation requires to integrate long range dependencies, making the PS-MNIST a reference benchmark to evaluate recurrent SNNs.

For both datasets, our models do not include any normalization layers, and training is performed without data augmentation. Our implementation builds upon the code of Xu et al.. For the SSC dataset, we used 3 fully connected hidden layers, with 256 neurons per layer, and we train our model on 3 different seeds. For the PS-MNIST, we use one layer of 64 neurons, then 2 layers of 212 neurons, and we only test one seed as all the previous state-of-the-art models on the dataset. Complete implementation details and hyperparameters are provided in A.2.5. Table 1 summarizes the accuracies of most competitive spiking, LIF-derived models on both datasets. We deliberately leave out of this table models that rely on substantially more complex neuron models, such as multi-compartment neurons (Zheng et al., 2024; Chen et al., 2024), attention or GRU based neurons (Dampfhoffer et al., 2022; Wang et al., 2024), whose additional mechanisms make direct comparison less meaningful[1].

Table 1: **Classification accuracy on SSC and PS-MNIST datasets**, ranked by accuracy.

| Model | Rec. | Rec. Delays | Ff. Delays | LIF | Param | Test Acc. [%] |
|---|---|---|---|---|---|---|
| **SSC** | | | | | | |
| Adaptive RSNN (Yin et al., 2021) | ✓ | | | | 0.78M | 74.20% |
| EventProp (Mészáros et al., 2025) | | | ✓ | ✓ | ∼ 5M[a] | 76.1±1.0% |
| RadLIF (Bittar & Garner, 2022) | ✓ | | | | 3.9M | 77.40% |
| cAdLIF (Deckers et al., 2024) | | | | ✓ | 0.35M | 77.60% |
| SE-adLIF (Baronig et al., 2025) | ✓ | | | | 1.6M | 80.44±0.26% |
| DCLS (Hammouamri et al., 2024) | | | ✓ | ✓ | 2.5M | 80.69±0.21% |
| ASRC-SNN (Xu et al.) | ✓ | ✓ | ✓ | | 0.37M | 81.54%* |
| SiLIF (Fabre et al., 2025) | ✓ | | | | 0.35M | 82.03±0.25% |
| **DelRec (Rec. and Ff. delays) *Ours*** | ✓ | ✓ | ✓ | ✓ | **0.55M** | **82.19±0.16%** |
| **DelRec (only Rec. delays) *Ours*** | ✓ | ✓ | | ✓ | **0.37M** | **82.58±0.08%** |
| **PS-MNIST** | | | | | | |
| GLIF (Yao et al., 2022) | ✓ | | | | 0.15M | 90.47% |
| Adaptive RSNN (Yin et al., 2021) | ✓ | | | | 0.15M | 94.30% |
| BRF (Higuchi et al., 2024) | ✓ | | | | 69k | 95.20% |
| ASRC-SNN (Xu et al.) | ✓ | ✓ | | ✓ | 0.15M | 95.77%* |
| **DelRec (only Rec. delays) *Ours*** | ✓ | ✓ | | ✓ | **0.16M** | **96.21%** |

[a] The parameter count is not clearly specified in the associated publication. Estimated from Figure 6.
[*] Results reproduced with publicly available code, using dedicated validation and test sets.

Overall, DelRec models set new state-of-the-art accuracy scores on both SSC and PS-MNIST datasets and using competitive numbers of parameters. Remarkably, this performance is achieved using simple LIF neurons, in contrast to many competing approaches that rely on more complex

---

[1]Dampfhoffer et al. (2022) report 77±0.4% on SSC using a spiking GRU, Zheng et al. (2024) report 82.46% on SSC (on only one seed) using multi-compartment neurons, Wang et al. (2024) report 83.69% on SSC (only one seed) using attention and distillation, Chen et al. (2024) report 97.78% on the PS-MNIST (one seed) with a multi-compartment neuron and using the test set as the validation set.

neuron models incorporating adaptive mechanisms, resonant dynamics, or structured state-space formulations. (Baronig et al., 2025; Fabre et al., 2025; Higuchi et al., 2024). This underscores the importance of incorporating synaptic delays in RSNNs (see also Hammouamri et al. (2024); Mészáros et al. (2025); Xu et al. for similar approaches) and suggests that even higher performance could be achieved by combining delays with more sophisticated neuron models. Our findings also demonstrate that optimizing synaptic delays can significantly enhance the performance of spiking models, even in recurrent networks. Furthermore, they indicate that optimizing delays in recurrent connections may yield greater benefits than optimizing feedforward delays, particularly for tasks involving long-range temporal dependencies (see Section 3.2).

## 3.2 FUNCTIONAL STUDY OF DELAYS

To determine whether the high accuracies in the previous section stemmed from the use of learnable recurrent delays, we compared our method with the state-of-the-art feedforward delay learning approach Hammouamri et al. (2024), and vanilla SNN architectures. It is worth noting that we are comparing synaptic feedforward delays (one delay per synapse), with axonal recurrent delays (one delay per neuron). For this study, we used much smaller models, and a much smaller dataset: SHD (Spiking Heidelberg Digits), a widely used spiking dataset (Cramer et al., 2022) of 10k recordings of spoken digits ranging from zero to nine, in English and German. This dataset demands leveraging temporal patterns in spike trains to reach good classification accuracies.

We adopted the following procedure. *Validation phase*: we first verified that our method performed competitively with feedforward delays on the SHD dataset. *Simplification phase*: we then reduced the size of the layers, and simplified our networks to obtain smaller versions of state-of-the-art models, with equivalent numbers of parameters. *Comparative phase*: we set the layer sizes such that each model contains around 10k parameters, then gradually reduced the number of neurons in layers, training a separate network for each model at each step. We compared the evolution of accuracies on the SHD as a function of the parameter count. We also enforced sparsity in the networks with spikes penalization, and study models' performance as a function of mean firing rate. Details of the architecture, parameters and training methodology can be found in A.2.3. In the next sections, we detail the results obtained during each if the phases described above.

***Validation phase***: To compete with state-of-the-art models on the SHD dataset, we designed a network inspired by Hammouamri et al. (2024), incorporating delays in recurrent connections. The architecture also included regularizing modules as described in Table 3. We augmented the training data following Mészáros et al. (2025) and Nowotny et al. (2022) (see A.2.3) to reduce overfitting. In all experiments on this dataset, we used networks with 2 hidden layers of 256 neurons, but recurrent connections (and recurrent delays) only in the second layer to avoid overfitting (Fig. 3A). As with the SSC and PS-MNIST datasets, we used only simple LIF neurons. However, we compared our architectural innovation to the best-performing models, which often rely on more sophisticated intrinsic neuronal dynamics.

SHD lacks a dedicated validation set, and historical evaluations of SNN performance on this dataset have relied solely on the test set. This approach is methodologically flawed and leads to an overfitting of the test set. More recent works have set more rigorous standards by using a fraction of the training set as a validation set, before reporting the best model's accuracy on the test set (Baronig et al., 2025; Mészáros et al., 2025). In line with this effort, we use $20\%$ of the non-augmented training set as a validation set, and we report the results we obtained on the test set in Table 2 (our models are trained on 10 different seeds). Yet, while the best models report around $93\%$ of accuracy on the test set (using a clean split), Mészáros et al. (2025) explain that further improvements in performance are likely not statistically significant given the small size of the test set (2264) : with naive assumptions on error rates, the Bayesian confidence intervals of accuracies over $93\%$ overlap. For all these reasons, we decided not to include SHD in Table 1.

Whether using both feedforward and recurrent delays or only recurrent delays, our models achieve state-of-the-art performance on SHD. Notably, the combination of recurrent and feedforward delays yields the highest mean accuracy among our tested configurations, demonstrating the effectiveness of integrating both delay types when overfitting is controlled (here with data augmentations). Given that our method improves the state-of-the-art on larger and more challenging datasets, it suggests that the SHD dataset has become overly saturated for benchmarking the processing capabilities of

Table 2: **Classification accuracies on SHD using a clean split**, ranked by accuracy.

| Model | Rec. | Rec. Delays | Ff. Delays | LIF | Param | Test Acc. [%] |
|---|---|---|---|---|---|---|
| BRF (Higuchi et al., 2024) | ✓ | | | | 0.1M | 92.70±0.70% |
| SE-adLIF (1L) (Baronig et al., 2025) | ✓ | | | | 37.5k | 93.18±0.74% |
| EventProp[b] (Mészáros et al., 2025) | | | ✓ | ✓ | ∼ 1M[a] | 93.24±1.00% |
| **DelRec (Only Rec. delays)** *Ours*[b] | ✓ | ✓ | | ✓ | **0.17M** | **93.39±0.45%** |
| **DelRec (Rec. and Ff. delays)** *Ours*[b] | ✓ | ✓ | ✓ | ✓ | **0.24M** | **93.73±0.69%** |
| DCLS[b] (Hammouamri et al., 2024) | | | ✓ | ✓ | 0.22M | 93.77±0.68% |
| SE-adLIF (2L) (Baronig et al., 2025) | ✓ | | | | 0.45M | 93.79±0.76% |

[a] The parameter count is not clearly specified in the associated publication. Estimated from Figure 5.
[b] Models trained using augmentations.

new spiking models. We therefore recommend its use only as an initial validation step for proof-of-concept studies.

***Simplification phase***: In Table 2, our configuration was exactly the same as the one in DCLS-Delays, and described in A.2.3. At this stage, we focused on simplifying both models to compare them at a lower yet equivalent number of parameters. The changes between the state-of-the-art models and the small models are detailed in Table 3.

Table 3: Network parameters for models used on the SHD dataset, ranked by accuracy.

| Model | #Layers | Hidden | $\tau$ (ms) | BN | Bias | Epochs | Augm. | #Params |
|---|---|---|---|---|---|---|---|---|
| Large | 2 | 256 | 10.05 | Yes | Yes | 150 | Yes | 0.2M |
| Small | 2 | ≤ 52 | 20 | No | No | 30 | No | ≤10k |

Additionally, we increased the learning rate to 0.1 and we applied a one cycle scheduler to the delay parameters. Feedforward delays were enabled only between the first and second hidden layers, while recurrent delays were restricted to the second hidden layer, ensuring that both types of delays operated on weighted spikes (see Fig. 3A).

***Comparative phase***: Recent works suggested that delays in SNNs improved robustness under low number of parameters and sparsity constraints (Hammouamri et al., 2024; Mészáros et al., 2025). To explore this further, we performed an ablation study on the SHD dataset, evaluating model performance under varying delay settings and constraints. In total, we compared 6 different models, with the architecture presented in Figure. 3A: a vanilla SNN, a vanilla RSNN with a uniform delay of 1 time-step in recurrent connections, a model with learned feedforward delays using DCLS-delays, a RSNN with fixed random delays in recurrent connections, a model with learned delays in recurrent connections and a model with learned feedforward and recurrent delays.

These studies highlight several observations regarding the role of delays. First, as theory suggests, models with either type of delays strongly outperform the equivalent architecture without delays (Fig. 3B), proving that delays, including those in recurrent connections, offer an invaluable tool for temporal structure extraction. Moreover, the comparison between a vanilla RSNN and the same network with random fixed recurrent delays illustrates how the simple introduction of delays in recurrent connections mitigates the training difficulties of RSNNs due to gradient issues.

Second, our results in Fig. 3C indicate that under low parameters constraints, a model with recurrent delays consistently outperforms all other models, with accuracy degrading less steeply as network size decreases. It suggests that recurrent delays allow for more efficient use and reuse of temporal information when representational capacity is limited. In contrast, we found no advantage in using both types of delays in these small configurations, despite this combination achieving our highest score on the SHD with larger models(Table. 2). However, Fig. 3 reveals a tradeoff between accuracy and energy consumption : while recurrent delays achieve better performance than feedforward delays for an equivalent number of parameters and without firing rate constraints, we found that feedforward delays reached their best accuracies with a lower mean firing rate than recurrent delays

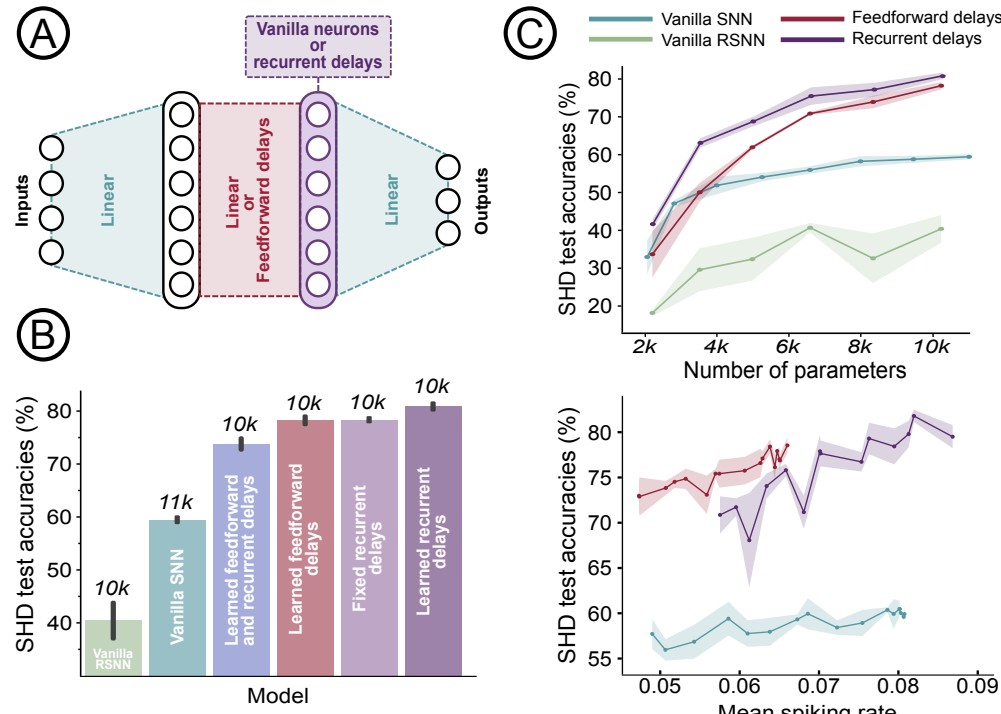

Figure 3: **A**: The generic architecture of networks used in this part. We use 2 hidden layers, with the inputs simply linearly mapped to the first layer, and the same for the readout. The linear mapping between the first and the second hidden layers can incorporate feedforward delays, and the neurons in the second layer can have recurrent connections, possibly with delays. **B**: Histogram of model accuracy on the SHD. The values on top of the bars are the number of parameters in the models. **C**: Model accuracy on the SHD as a function of the number of parameters in the network (*top*), and as a function of the mean number of spikes per neuron per time-step (*bottom*). The shaded areas represent the standard error mean (sem). For more details see A.3.

required for the same performance. Though the model with recurrent delays achieves its best accuracy under an already low energy cost (0.08 spikes per neuron per time-step), our study suggest that feedforward delays can provide a more energy-efficient alternative when computational efficiency is preferred over performance.

Finally, in line with the findings of Hammouamri et al. (2024) with DCLS-Delays, we observe in Fig. 3B that the benefit of learning delays in recurrent connections is relatively small, yet consistent and significant, underlining the utility of optimizing recurrent dynamics for temporal processing.

## 4 CONCLUSION

This work introduces a new method (DelRec) to optimize delays in the recurrent connections of spiking neural networks with surrogate gradient learning and backpropagation, leveraging differentiable interpolation and a progressive spike scheduling process. Using the simplest spiking neuron model, i.e., a LIF with instantaneous synapses, DelRec outperforms the previous state-of-the-art accuracy on both the PS-MNIST vision dataset and the SSC audio dataset, two widely recognized benchmarks for evaluating temporal processing capabilities. Moreover, we present a study suggesting that recurrent delays can achieve better performance than feedforward delays. We believe that further improvements could be obtained by using more complex neurons with DelRec, and by better combining DelRec with feedforward delays. Finally, our method also offers new tools for modeling neural populations dynamics in the brain and could offer insights on how delays shape sensory processing.

## 5 REPRODUCIBILITY STATEMENT

All the results presented in this work can be reproduced using the anonymous repository: `https://anonymous.4open.science/r/Recdel-4175`. We used publicly available datasets, downloadable at the following address: `https://zenkelab.org/datasets/`. We also used the PS-MNIST dataset, which directly derived from Pytorch's MNIST. Our implementation builds upon the Spiking Jelly framework (Fang et al., 2023), an open-source library providing optimized tools for developing spiking neural networks. The hyperparameters we used are provided in the Appendix (see A.2.5), and can also be found in the configuration files of our repository. Finally, our results were produced using NVIDIA A100 GPUs for the SSC and PS-MNIST datasets, and NVIDIA A40 GPUs for the SHD dataset.

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

# A  APPENDIX

## A.1  LEARNING ALGORITHM

We detail here the algorithm we use to learn delays in recurrent connections. Algorithm 1 describes a forward pass for $N$ neurons. Our spike scheduling employs a circular buffer of size $N \times \dim(\tilde{\mathbf{E}}(\sigma, D))$ with $\tilde{\mathbf{E}}(\sigma, D)$ defined in Eq. 13.

## A.2  LEARNING STRATEGY, ARCHITECTURES AND HYPERPARAMETERS

### A.2.1  $h_{\sigma, d}$

Our algorithm uses a scheduling of spikes for future time steps, which is, for one recurrent connection, spread around the delay value $d$. This spread has a triangular shape determined by $h_{\sigma, d}$, with the sigma decreasing from its initial value to zero throughout the training. We use an exponential decay value $decay = 0.95$, and we update $\sigma_{epoch}$ at each epoch such that :

$$\sigma_{epoch} \leftarrow \sigma_{init} \times decay^{100 \times \frac{epoch}{N_{\text{epochs}}}} \tag{14}$$

with $N_{\text{epochs}}$ the total number of epochs. We always take $\sigma_{init} = 10$.

However, we use an additional strategy on the SSC dataset in order to allow for a quicker decay of $\sigma_{epoch}$ for specific neurons. Specifically, we introduce a parameter $p$ per neuron, and consider for a neuron $i$ the modified spread function:

$$h_{\sigma, d_i, p_i}(\tau) = \max(0, \frac{1 + 2 \cdot \text{sig}(p_i) \cdot \sigma - |\tau - (1 + d_i)|}{(1 + 2 \cdot \text{sig}(p_i) \cdot \sigma)^2}) \tag{15}$$

with $\text{sig}(\cdot)$ the sigmoid function, and parameters $\{p_i\}$ all initialized at 0, and added to the learnable parameters. To remain neuromorphic hardware friendly and have a single integer delay in each recurrent connection at the end of training, the $\sigma$ parameter is still updated with Eq. 14.

### A.2.2  SSC AND PS-MNIST DATASETS

On the SSC dataset, we reduced the input size by binning every 5 neurons of the original 700, resulting in a spatial input dimension of 140. We also temporally binned the inputs using a discrete time-step of $\Delta = 5.6$ms.

---

**Algorithm 1:** Learning Delays in Recurrent Connections

---

**Inputs:** Sequence time point at date $t$: $X[t]$, for $N$ neurons.
**Parameters:** Number of neurons $N$, recurrent weights $W^{\text{rec}} \in \mathbb{R}^{N \times N}$, recurrent delays $d \in \mathbb{R}^N$,
         scheduled $\sigma_{epoch}$, firing threshold $V_{\text{th}}$, neuron reset type, reset potential $V_{reset}$, neuronal
         charge function $f$.
**Outputs:** Spikes at time $t$: $S[t]$.
Compute $\tilde{\mathbf{E}}(\sigma_{epoch}, D)$
$L \leftarrow \dim\big(\tilde{\mathbf{E}}(\sigma_{epoch}, D)\big)$
$\texttt{spread} \leftarrow \big[h_{\sigma_{epoch}, d}(x) \text{ for } x \in \tilde{\mathbf{E}}(\sigma_{epoch}, D)\big]$
$B \leftarrow \mathbf{0}^{N \times L}$                                 `// zero-out buffer matrix`
$\texttt{pointer} \leftarrow 0$
**for** $t \leftarrow 0$ **to** $T$ **do**
    **for** $i \leftarrow 1$ **to** $N$ **do**
        $X_i^{\text{rec}}[t] \leftarrow B_i[\texttt{pointer}]$         `// take current time step in the buffer`
        $I_i[t] \leftarrow X_i[t] + X_i^{\text{rec}}[t]$       `// get current spikes with neuron equation`
        $H_i[t] \leftarrow f(V_i[t-1], I_i[t])$
        $S[t] \leftarrow \Theta\big(H_i[t] - V_{\text{th}}\big)$
        **if** *Hard reset* **then**
            $V_i[t] \leftarrow H_i[t] \cdot (1 - S_i[t]) + V_{\text{reset}} \cdot S_i[t]$
        **else**
            $V_i[t] \leftarrow H_i[t] - V_{\text{th}} \cdot S_i[t]$
        **end**
        $B_i[\texttt{pointer}] \leftarrow 0$                         `// update the buffer`
        $\texttt{pointer} \leftarrow (\texttt{pointer} + 1) \bmod L$
        **for** $j \leftarrow 1$ **to** $N$ **do**
            **for** $\tau \in \tilde{\mathbf{E}}(\sigma_{epoch}, D)$ **do**
                **if** $\texttt{pointer} + \tau \leq L - 1$ **then**
                    $B_j[\texttt{pointer} + \tau] \leftarrow B_j[\texttt{pointer} + \tau] + w_{ji}^{rec} \cdot \texttt{spread}[\tau] \cdot S_i[t]$
                **else**
                    $\texttt{p}' \leftarrow (\texttt{pointer} + \tau) \bmod L$
                    $B_j[\texttt{p}'] \leftarrow B_j[\texttt{p}'] + w_{ji}^{rec} \cdot \texttt{spread}[\tau] \cdot S_i[t]$
                **end**
            **end**
        **end**
    **end**
**end**

---

For both the SSC and PS-MNIST, the readout is a simple linear layer mapping the last hidden layer to the $n_{\text{classes}} = 35$ output neurons. Then, denoting $N$ the batch size, $\hat{y}_i$ the output of neuron $i$ and $y$ the ground truth, we compute the cross-entropy loss for one batch as:

$$\mathcal{L} = \frac{1}{N} \sum_{n=1}^{N} -\log(\hat{y}_{n,\, y_n}[n]). \tag{16}$$

### A.2.3 SHD DATASET

For this dataset, we also reduced the input size by binning every 5 neurons of the original 700, resulting in a spatial input dimension of 140. As well, we binned the inputs temporally, using a discrete time-step of $\Delta = 10$ms. The architecture used is composed of two fully connected hidden layers (see Fig. 3.A), and each of these layers can be followed by batch normalization, a neuron module and dropout. The readout consists of $n_{\text{classes}} = 20$ LIF neurons with an infinite threshold. Following Bittar & Garner (2022), we consider $V_i^r$ the membrane potential of the readout neuron $i$, then we compute the softmax over output neurons at each time-step before summing across the temporal dimension. Thus, the model output for this neuron:

$$\hat{y}_i = \sum_{t \in T} \text{softmax}(V_i^r[t]) = \sum_{t \in T} \frac{e^{V_i^r[t]}}{\sum_{j=1}^{n_{\text{classes}}} e^{V_j^r[t]}} \tag{17}$$

with $T$ the temporal dimension. Then, denoting $N$ the batch size and $y$ the ground truth, we compute the cross-entropy loss for one batch as:

$$\mathcal{L} = \frac{1}{N} \sum_{n=1}^{N} -\log(\hat{y}_{n,\, y_n}[n]). \tag{18}$$

We used augmentations on the training set of the SHD in order to limit overfitting. We use this method only on our large models when trying to reach high accuracies, not in our ablation study. We followed the approach of Nowotny et al. (2022) and Mészáros et al. (2025):

- We applied a random temporal shifting to inputs, with a shift uniformely drawn in the interval $\big[-100 \text{ time steps},\ 100 \text{ time steps}\big]$.

- Each input was blended with an input from the same class by aligning their center of mass and, for each time step, picking a spike from either sample with a probability of $0.5$.

### A.2.4 WEIGHTS AND DELAYS INITIALIZATION

**SSC and PS-MNIST**: We initialized the feedforward weights with the default linear weights and biases method in Pytorch:

$$w_{ij}, b_i \sim \mathcal{U}\left(-\sqrt{\tfrac{1}{\text{in\_features}}},\ +\sqrt{\tfrac{1}{\text{in\_features}}}\right).$$

The recurrent delays were initialized with a random draw from a half gaussian centered on $0$:

$$d_i^{\text{rec}} \sim |\mathcal{N}(0, \sigma^2)|,\ \text{with } \sigma = 12.$$

**SHD**: We initialized the feedforward weights using Kaiming uniform initialization, with $a = \sqrt{5}$:

$$w_{ij} \sim \mathcal{U}\left(-\sqrt{\tfrac{6}{\text{in\_features}}},\ +\sqrt{\tfrac{6}{\text{in\_features}}}\right),$$

$$b_i \sim \mathcal{U}\left(-\tfrac{1}{\sqrt{\text{in\_features}}},\ +\tfrac{1}{\sqrt{\text{in\_features}}}\right),$$

where fan_in denotes the number of input units to the layer. The recurrent delays were initialized with a random draw from a uniform distribution:

$$d_i^{\text{rec}} \sim \mathcal{U}(10,\ 30).$$

For all tasks, in the models with feedforward delays, we initialized them uniformly with :

$$d_i^{\text{ff}} \sim \mathcal{U}(0,\ 50).$$

### A.2.5 HYPERPARAMETERS

We detail here all the hyperparameters that we used to perform the experiments presented in this work.

In our models, the feedforward delays were optimized with the DCLS method. However, while Hammouamri et al. (2024) use gaussians centered on the delays positions in their temporal kernels and reduce the variance during training, we only spread the weights using a simple linear interpolation between the two closest integer positions to the floating delay parameter.

On the SSC dataset, we changed the feedforward dropout to 0.2 when feedforward delays were used, in order to avoid overfitting.

Table 4: General hyperparameters used for the SSC, PS-MNIST and SHD datasets.

| Hyperparameter | SSC | PS-MNIST | SHD |
|---|---|---|---|
| epochs | 100 | 200 | 150 |
| batch size | 128 | 256 | 256 |
| learning rate max weights | 1e-3 | 1e-3 | 5e-3 |
| learning rate max positions | 5e-2 | 5e-2 | 5e-2 |
| optimizer | Adam | AdamW | AdamW |
| weight decay | 1e-5 | 1e-2 | 1e-5 |
| batch norm | False | False | True |
| bias | False | False | True |
| feedforward dropout | 0.1 | 0.1 | 0.4 |
| recurrent dropout | 0.3 | 0.2 | 0.2 |
| scheduler weights | One cycle | One cycle | One cycle |
| scheduler positions | One cycle | One cycle | Cosine annealing |

Table 5: Neuron-related hyperparameters used for the SSC, PS-MNIST and SHD datasets.

| Hyperparameter | SSC | PS-MNIST | SHD |
|---|---|---|---|
| $\tau$ (time-steps) | 2 | 2 | 1.005 |
| $V_{threshold}$ | 1. | 1. | 1. |
| reset type | Soft | Soft | Hard |
| $V_{reset}$ | | | 0 |
| detach reset | False | False | True |
| surrogate function | Triangle | Triangle | Arctan |

## A.3 ABLATION STUDY METHODOLOGY

In this section, we aim to explain in more details our methodology to produce Fig. 3.

Fig 3B : The goal was to obtain acceptable accuracies with a minimal number of parameters. We found that around 10k parameters was an overall effective choice. To this end, we picked the number of neurons reported in Table 6 for our different models, and we trained them on 3 seeds. The standard error mean (sem) is also reported on the figure.

Table 6: Neurons per layer in the models of the ablation study.

| Model | Hidden layers sizes | Approx. number of parameters |
|---|---|---|
| Vanilla RSNN | [42, 42] | $\sim$ 10k |
| Vanilla SNN | [52, 52] | $\sim$ 11k |
| Learned Ff. and Rec. delays | [38, 38] | $\sim$ 10k |
| Learned Ff. delays | [42, 42] | $\sim$ 10k |
| Fixed Rec. delays | [42, 42] | $\sim$ 10k |
| Learned Rec. delays | [42, 42] | $\sim$ 10k |

Fig 3C, *Top*: For each model, we started with the same layer size reported in Table. 6, and we reduced gradually the layer sizes with steps of 6 neurons, training on 3 seeds each time, to produce the points in the figure.

Fig 3C, *Bottom*: We started with the same models as in Table. 6, but we gradually penalized the mean firing rate of the model in order to introduce more sparsity. To do so, we added the following term to the loss, as in Pellegrini et al. (2021) :

$$\mathcal{L}_{\text{spike}} = \frac{1}{2TBN} \sum_{b=1}^{B} \sum_{t=1}^{T} \sum_{n=1}^{N} S_{b,n}[t]^2,$$

where $T$ is the length of the temporal dimension, $N$ is the number of neurons in the model and $B$ is the batch size. Thus, we considered a total loss:

$$\mathcal{L} = \mathcal{L}_{\text{cross entropy}} + \lambda \mathcal{L}_{\text{spike}},$$

and we trained our models (3 seeds again) for $\lambda$ values ranging from 1e-4 to 10. We recorded the number of spikes per neuron, and we plotted the model accuracy as a function of the number of spike per neuron and per time-step on the test set.

## A.4 USE OF LARGE LANGUAGE MODELS

We declare the following use of LLMs in the preparation of this work:

- For writing: OpenAI's ChatGPT was used in the writing of this article to improve the phrasing and the clarity of some sentences.
- For coding: OpenAI's ChatGPT and GitHub copilot were used to enhance coding efficiency, more particularly for the factorization of existing scripts and the generation of plots.

