# OpenReview forum: "DelRec: learning delays in recurrent spiking neural networks."
_ICLR.cc/2026/Conference — ICLR 2026 Conference Withdrawn Submission_

### Official Review · Reviewer_MFAp · 2025-10-31

**Soundness:** 1
**Presentation:** 1
**Contribution:** 1
**Rating:** 0
**Confidence:** 4

**Summary:**

This paper proposes DelRec that allows recurrent axonal delays to be learnable parameters. Given the use of real value delays, the conventional backprop can be used together with gradual quantization of delays. The proposed method was applied to several sequential data like SHD, SSC, PS-MNIST.

**Strengths:**

Propose of novel delay quantization algorithm: the authors propose a novel algorithm for gradual delay quantization.

**Weaknesses:**

Limited novelty: A very similar method was proposed back in 2018 (see SLAYER in NeurIPS 2018). SLAYER is an SNN-training framework based on error-backprop with SGL. The only difference of the proposed work with SLAYER lies in its application to recurrent-type SNNs and delay quantization.

Weak evaluation of delay quantization: it is not clear the value of the delay quantization method since it was not analyzed in the present manuscript.

Unclear scalability: the proposed method was applied to only small datasets.

Unfair comparison: the accuracy of the proposed method on the datasets was not compared with SoTA accuracy. I found much higher accuracy than those used in the comparison. Further, DCLS’s accuracy in Table 2 should be 95.07 rather than 93.77.

**Questions:**

The authors should explain why some benchmark accuracy reported in this manuscript is different from the real one.

---

> ### Author Response · Authors · 2025-11-18
>
> Thank you for your thoughtful review. We address all the listed weaknesses in the following.
>
> 1) "Limited novelty: A very similar method was proposed back in 2018 (see SLAYER in NeurIPS 2018). SLAYER is an SNN-training framework based on error-backprop with SGL. The only difference of the proposed work with SLAYER lies in its application to recurrent-type SNNs and delay quantization."
>
> SLAYER, which we cited, is indeed a pioneering work: it was the first method to train delays with SGL. However, SLAYER is limited to feedforward networks, and another limitation is the restriction to the spike response model. Since then, broader architectures have emerged, complex neurons have been developed and much more challenging spatio-temporal datasets have been commonly used to benchmark long-range capabilities of SNNs. SLAYER led to development of many modern works that are more general, more powerful and more scalable : [1, 2, 3, 4]. All these works enabled new performances via better delay learning, yet none of them extend their work to recurrent delays, indicating that the extension is not trivial. Very recently, [5] proposed a first method to learn delays in recurrent connections in continuous time and by exact gradient computation. On the other hand, we introduce the first method to train delays in recurrent SNNs with SGL, which is in addition compatible with most spiking neuron models. This method led to new SOTAs on the main benchmarks used in the SNN community for temporal processing, showed an advantage of delays in recurrent connections over delays in feedforward connections on the datasets we used, and resolves an unaddressed problem in the SNN community. We believed that it was a bit more than an extension of SLAYER, and that it was worth sharing.
>
> [1] J. Goltz et al. Delgrad: exact event-based gradients for training delays and weights on spiking neuromorphic hardware. Nature Communications, 2025.
> [2] I. Hammouamri et al. Learning Delays in Spiking Neural Networks using Dilated Convolutions with Learnable Spacings. In ICLR, pp. 1–12, 2024.
> [3] Deckers et al. Co-learning synaptic delays, weights and adaptation in spiking neural networks. Frontiers in Neuroscience, 2024.
> [4] M. Ghosh et al. Spiking Neural Network Models of
> Interaural Time Difference Extraction via a Massively Collaborative Process. eneuro, 2025.
> [5] Balazs Mészaros, James C. Knight, and Thomas Nowotny. Efficient event-based delay learning in spiking neural networks, 2025.
>
> 2) "Weak evaluation of delay quantization: it is not clear the value of the delay quantization method since it was not analyzed in the present manuscript."
>
> We thank the reviewer for the opportunity to clarify this point: the delays are rounded at the end of each epoch throughout training, which leads to a convergence extremely close to integer delays. We observed that doing this is improved the training of our models similarly to adding a small temporal jitter, as well leading to a more natural convergence to integer delays. After running a test on 10 seeds on SHD with our best model, we get a score of $93.23 \pm 0.28$ when we don't round neither during training and testing, compared to $93.39 \pm 0.45$ when rounding at each epoch.
>
> 3) "Unclear scalability: the proposed method was applied to only small datasets."
>
> This claim is very subjective and biased: we applied our method to the two datasets that are used in the SNN community to benchmark spatio-temporal processing, the PS-MNIST (Vision) and the SSC dataset. The SSC has over 100K audio samples distributed over 35 classes of spoken commands. It is one of the biggest and the most challenging dataset used in the community. We also applied our method to the PS-MNIST dataset, which contains 60,000 images that we flatten into sequences of length 784 and permute the pixels positions. This dataset is also widely used to benchmark the integration of long range dependencies. These two datasets, SSC and PS-MNIST, are standard, widely accepted benchmarks for evaluating spatio-temporal and sequential processing in SNNs. We would like to stress out that the performances we achieved in our work were obtained with LIF neurons, one of the most simple neuron models, indicating that further improvements should arise when recurrent delays will be combined with relevant more complex and expressive neuron models.

---

> ### Author Response · Authors · 2025-11-18
>
> 4) "Unfair comparison: the accuracy of the proposed method on the datasets was not compared with SOTA accuracy. I found much higher accuracy than those used in the comparison. Further, DCLS’s accuracy in Table 2 should be 95.07 rather than 93.77."
>
> We will quote our paragraph in section 3.2 to answer this question: "SHD lacks a dedicated validation set, and historical evaluations of SNN performance on this dataset have relied solely on the test set. This approach is methodologically flawed and leads to an overfitting of the test set. More recent works have set more rigorous standards by using a fraction of the training set as a validation set, before reporting the best model’s accuracy on the test set (Baronig et al., 2025; Mészaros et al., 2025). In line with this effort, we use 20$\%$ of the non-augmented training set as a validation set, and we report the results we obtained on the test set in Table 2 (our models are trained on 10 different seeds)". In other words, we reproduced the DCLS result with their accessible code using a clean methodology with a validation set separated from the test set. All the model reported in Table 2 report results obtained using this methodology. Furthermore, as it is implied by the sentence "Models trained using augmentations." below the Table 2, the same augmentations that we used were applied to the original DCLS method in order to have a clearer performance benchmark between feedforward and recurrent delays, using an exactly similar methodology. This fair methodology in fact increased the performance of DCLS on this dataset.

---

### Official Review · Reviewer_rGry · 2025-11-01

**Soundness:** 2
**Presentation:** 2
**Contribution:** 2
**Rating:** 2
**Confidence:** 5

**Summary:**

The authors aim to improve the performance of recurrent spiking neural networks by introducing tunable axonal/synaptic delays. This work leverages the standard BP with surrogate gradients to train such networks.

**Strengths:**

**Pros**：

- The paper looks into learning delays in recurrent spiking networks, which has not be extensively explored in the past.

- The paper is well written, easy to follow while some of the key ideas should be better motivated.

- Only pretty modest performance gain is demonstrated.

**Weaknesses:**

- In general the notion of learnable delays is well understood although this was mostly explored in feedforward networks. Extending it for recurrent networks is relatively stratforward.

- The proposed technique for handling learnable delays in recurrent spiking nets doesn't seem to have much novelty. This only involves with some relatively minor technical handling.

- The main point of the paper centers around (9) where a targeted real-valued delay gets mapped to a set of integer delay values with a width parameter $\sigma$. Why is this needed? Is the main motivation just to smooth out the discreteness?

**Questions:**

- Around (12), it is mentioned, based on a real-valued delay, the recurrent inputs are "scheduled" within a limited range of time steps? What do you mean by "scheduling"? Do you approximate the real-valued delay by randomly selecting an integer delay value in the range? What's the impact of doing this on training?

- It is also mentioned that in the end, the support of the integer delays reduces to two adjacent time points， and  rounding to the nearest integer delay is performed for inference. What is the accuracy impact of doing this?

---

> ### Author Response · Authors · 2025-11-18
>
> Dear reviewer, we thank you for your review. We will address the weaknesses you point out, then your questions.
>
> 1)"In general the notion of learnable delays is well understood although this was mostly explored in feedforward networks. Extending it for recurrent networks is relatively straightforward." & "The proposed technique for handling learnable delays in recurrent spiking nets doesn't seem to have much novelty. This only involves with some relatively minor technical handling."
>
> This is a very subjective claim and does not accurately reflect the state of the literature. A more objective summary of prior work on delay learning in Spiking Neural Networks is as follows:.
> - Delay learning in SNNs has been studied almost exclusively on feedforward SNNs [1, 2, 3, 4, 5], yielding significant advances in spatiotemporal processing in SNNs. Yet, none of the mentioned works extend their models to recurrent architectures, which are the theoretical optimum for long-range temporal tasks, indicating that the extension is not so straightforward.
> - An obstacle to this extension probably lies in how spiking layers are coded, because we usually run the whole time loop separately for the different layers of a SNN. This allows to add delays directly between layers, as [3] for example, who use a convolution between layers to delay the spike trains. However, delaying spikes in recurrent connections implies to modify the internal time loop of each layer and scheduling spikes for next time steps in the loop, while making it differentiable, efficient and competitive. We found it to be non-trivial.
> - Only one work before us, which is also very recent, tackled the tasks of optimizing delays in recurrent connections of SNNs [6] in the framework of the Eventprop algorithm that leverages the computation of exact gradients. In our case, we propose an algorithm to perform this learning is using surrogate gradients, the method used by all SOTA SNN models on temporal benchmarks. Our method relies on a per neuron recurrent delay learning with a differentiable scheduling over an optimal support that did not exist before, and brings a first response to an unsolved problem in the SNN community. Moreover, we also performed a comparison of learned feedforward and recurrent delays under multiple relevant architectural and computational constraints, and this analysis revealed that trainable recurrent delays could have some advantages over feedforward ones. To the best of our knowledge, that kind of analysis did not appear before, and provides new insights that are directly relevant for model design, biological modeling and neuromorphic deployment.
>
> [1] Shrestha et al. Slayer: Spike layer error reassignment in time, 2018.
> [2] J. Goltz et al. Delgrad: exact event-based gradients for training delays and weights on spiking neuromorphic hardware. Nature Communications, 2025.
> [3] I. Hammouamri et al. Learning Delays in Spiking Neural Networks using Dilated Convolutions with Learnable Spacings. In ICLR, pp. 1–12, 2024.
> [4] Deckers et al. Co-learning synaptic delays, weights and adaptation in spiking neural networks. Frontiers in Neuroscience, 2024.
> [5] M. Ghosh et al. Spiking Neural Network Models of
> Interaural Time Difference Extraction via a Massively Collaborative Process. eneuro, 2025.
> [6] Balazs Mészaros, James C. Knight, and Thomas Nowotny. Efficient event-based delay learning in spiking neural networks, 2025.
>
> 2) "The main point of the paper centers around (9) where a targeted real-valued delay gets mapped to a set of integer delay values with a width parameter $\sigma$. Why is this needed? Is the main motivation just to smooth out the discreteness?"
>
> Yes, this is the main motivation: by scheduling spikes to a broader range of time steps at the beginning of training, we allow the model to capture a longer range of events than on a simple time step. The smooth decrease of the sigma throughout the training facilitates the adaptation of the weights and delay parameters to a tighter scheduling over a reduced set of time steps.
>
> 3) "Around (12), it is mentioned, based on a real-valued delay, the recurrent inputs are "scheduled" within a limited range of time steps? What do you mean by "scheduling"? Do you approximate the real-valued delay by randomly selecting an integer delay value in the range? What's the impact of doing this on training?"
>
> Our method does not involve any random process. In our paper, scheduling refers to saving emitted spikes at one time step, and deterministically distributing them to future time steps. We smooth out this process as you mentioned, by using a triangular interpolation kernel centered at the learned real-valued delay, with a width that is reduced to zero by the end of training. During training, this smooth interpolation avoids discontinuous jumps that would make delay learning unstable. At inference, delays are rounded to their nearest integer values.

---

> ### Author Response · Authors · 2025-11-18
>
> 4) "It is also mentioned that in the end, the support of the integer delays reduces to two adjacent time points， and rounding to the nearest integer delay is performed for inference. What is the accuracy impact of doing this?"
>
> Thank you for the opportunity to clarify this: the delays are rounded at the end of each epoch throughout training, which leads to a convergence extremely close to integer delays. We observed that doing this is improved the training of our models similarly to adding a small temporal jitter, as well leading to a more natural convergence to integer delays. After running a test on 10 seeds on SHD with our best model, we get a score of $93.23 \pm 0.28$ when we don't round neither during training and testing, compared to $93.39 \pm 0.45$ when rounding at each epoch.

---

### Official Review · Reviewer_izGg · 2025-11-05

**Soundness:** 2
**Presentation:** 2
**Contribution:** 2
**Rating:** 2
**Confidence:** 4

**Summary:**

This paper introduces DelRec, a novel method for learning neuron-level recurrent connection delays in SNNs. Its core idea is to use differentiable interpolation techniques to transform the discrete delay optimization problem into a continuous gradient optimization problem, thereby enabling end-to-end training within the mainstream Surrogate Gradient Learning (SGL) framework.

**Strengths:**

The core contribution of the paper is its successful integration of learnable recurrent delays into the SGL framework, achieving state-of-the-art (SOTA) performance on two recognized SNN temporal benchmarks (SSC and PS-MNIST) using only simple LIF neurons. And the ablation study on the SHD dataset (Figure 3) provides valuable insights into the performance and efficiency trade-offs of different delay mechanisms (feedforward vs. recurrent).

**Weaknesses:**

1. As shown in Table 1, the authors' reproduction results for ASRC-SNN significantly differ from the original paper (the original paper achieved 96.62% on PS-MNIST with 0.15M parameters, while this paper's reproduction is only 95.77%). Moreover, their arbitrary change of configuration for different datasets (e.g., using LIF for PS-MNIST but adding Ff. delays for SSC) has unclear motivations. This severely undermines the credibility of their SOTA claim.

2. The DelRec method uses a larger $\alpha$ value during initial training, which means a spike needs to be scheduled within a wide time window (Equation 12). This may incur significant memory and computational overhead. The paper completely lacks analysis of its method's spatial/temporal complexity in relation to maximum delay, nor does it provide any reports on training/inference time consumption. This makes it impossible to judge whether the method is feasible for processing large datasets requiring long-range dependencies (such as EEG signals) or on resource-constrained hardware.

3. The experimental results of the paper show internal contradictions. On the SSC dataset (Table 1), the "recurrent delay only" model performs better than the "recurrent and feedforward delay combined" model. However, on the SHD dataset (Table 2), the authors claim that the combination of both works best. The authors do not provide any explanation for this inconsistency, which casts doubt on the reliability and generality of their conclusions.

4. While learning recurrent delays within the SGL framework might be new, the idea of "learning connection delays" has long existed in the broader field of RNNs. The paper needs to discuss early related works (missing the related works section). Furthermore, the authors claim in the abstract that their work "paves the way for neuromorphic hardware deployment," but the entire paper provides no discussion or evidence related to hardware, which overstates the contribution.

5. There are multiple instances of unclear symbols and formula definitions in the paper. For example, $supp(h_\alpha,d)$ in Equation (12) and Rec. Delays in Table 1 and 2 are undefined in the main text, seriously affecting the readability and reproducibility of the paper.

**Questions:**

1.Please provide a detailed explanation of the complete hyperparameters used when reproducing ASRC-SNN. Why is there a discrepancy between your reproduction results and the original paper? Also, why are different network structure configurations used for ASRC-SNN on the SSC and PS-MNIST datasets?

2.I recommend that you supplement your paper with an analysis of the time and memory complexity of the DelRec method. Specifically, how does its overhead scale with maximum delay and sequence length?

3.Can you comment on the feasibility of this method for processing larger-scale temporal datasets (such as EEG)?

---

> ### Author Response · Authors · 2025-11-18
>
> Dear reviewer, thank you for your review. We will address all the weaknesses, which include your two questions.
>
> 1) "As shown in Table 1, the authors' reproduction results for ASRC-SNN significantly differ from the original paper (the original paper achieved 96.62\% on PS-MNIST with 0.15M parameters, while this paper's reproduction is only 95.77\%).
>
> You are right about the difference of scores: the original ASRC-SNN paper reports scores of 81.93 on SSC and 96.62 on PS-MNIST. However, this work uses the test set of each dataset as a validation set. The SSC dataset provides a dedicated validation set, and we believe that the rigorous methodology is to use a validation set during training and evaluate our trained model on the test set only once at the end of training. Therefore, we used the code provided by the authors of ASRC-SNN to test their method with dedicated validation and test sets on the SSC dataset, and we did the same on PS-MNIST by taking 50,000 of the 60,000 training samples for training, and 10,000 for validation. We agree that the methodology could be more carefully described than by our actual sentence: "Results reproduced with publicly available code, using dedicated validation and test sets", for example by adding a subsection in the appendix on this topic.
>
> 2) "Moreover, their arbitrary change of configuration for different datasets (e.g., using LIF for PS-MNIST but adding Ff. delays for SSC) has unclear motivations. This severely undermines the credibility of their SOTA claim."
>
> We do not understand this weakness: The LIF neuron was used *for all experiments*, and we tried to combine recurrent and feedforward delays on the SSC dataset, which has not been fruitful. This was consistent with all our experiments, where we saw that the combination of both types of delays in our framework did not improve significantly the performance when compared to recurrent delays only. Therefore, we did not try the combination of both on the PS-MNIST dataset. Still, our best scores on both datasets were achieved with recurrent delays only, the methodology and hyperparameters of our experiments are detailed in the appendix, and more importantly the code to reproduce and verify the results is provided with the paper. We believe that this last point is the most we can do to ensure the credibility of our SOTA claims.
>
> 3) "The DelRec method uses a larger $\alpha$ value during initial training, which means a spike needs to be scheduled within a wide time window (Equation 12). This may incur significant memory and computational overhead."
>
> We indeed use a triangle function to spread the future values around the real delay value $d$, with a $\sigma$ decreasing with training. At each forward pass, and buffer is created to store future delay values, and the size of this buffer is Neurons $\times$ $1 + \max d_i + (1+\sigma)$. In all our experiments, the $\sigma$ used is at maximum $10$ time-steps at the beginning our training (with sequences of around 100 time steps for SSC and 784 for PS-MNIST). Therefore, is only increases the scheduling buffers size by at maximum 10, which is a negligible memory cost compared to using a $\sigma = 0$.
>
> 4) "The paper completely lacks analysis of its method's spatial/temporal complexity in relation to maximum delay, nor does it provide any reports on training/inference time consumption. This makes it impossible to judge whether the method is feasible for processing large datasets requiring long-range dependencies (such as EEG signals) or on resource-constrained hardware."
>
> Thank you for this question, this is indeed an improvable point of the paper. During training, a forward pass in a regular Recurrent Spiking Neural Network (RSNN) layer has a temporal complexity of $O(BS)$ per time-step, with $B$ the inputs batch size and $S = N_{in}N_{out}$ the number of synapses of the recurrent connections. When using DelRec, the time-complexity increases to $O(BSL)$ with $L$ the effective delay buffer length, of the order of $\max d_i$. The same logic applies for the extra memory: a regular RSNN as a memory cost of $O(BN)$, with $N$ the number of neurons in the layer, and the DelRec increases this cost to $O(BNL)$. We will add a section focused on measuring the computational overhead of our method over vanilla training on our datasets. For preliminary runs with our best model on the SHD dataset, using a NVIDIA A100 GPU and with the hyperparameters detailed in the supplementary of the papers, DelRec leads to an increase of computational time of approximately 18\% compared to a vanilla RSNN.

---

> ### Author Response · Authors · 2025-11-18
>
> 5) "The experimental results of the paper show internal contradictions. On the SSC dataset (Table 1), the "recurrent delay only" model performs better than the "recurrent and feedforward delay combined" model. However, on the SHD dataset (Table 2), the authors claim that the combination of both works best. The authors do not provide any explanation for this inconsistency, which casts doubt on the reliability and generality of their conclusions."
>
> On all datasets, the model with recurrent delays only performs significantly better than the models with both types of delays, except in our experiments with SOTA models in SHD. Yet, in this last experiment, the difference between the model with both types of delays and the one with only recurrent delays is not significant. We do not have a clear theory on why that is the case, apart from the over parametrization of the networks when using the two types of delays, which results in overfitting. Still, most SNNs works show data dependent behavior, which is inherently linked to the temporal structure of the datasets. We demonstrate strong on consistent performance over multiple challenging datasets with very competitive number of parameters, and our conclusions on smaller networks are shared by [1], which is the only other work existing on learning delays in RSNNs. The claim that it 'casts doubt on the reliability and generality of [our] conclusions' feels like an heavy and unfounded exaggeration.
>
> [1] Balazs Mészaros, James C. Knight, and Thomas Nowotny. Efficient event-based delay learning in spiking neural networks, 2025. URL https://arxiv.org/abs/2501.07331.
>
> 6) "While learning recurrent delays within the SGL framework might be new, the idea of "learning connection delays" has long existed in the broader field of RNNs. The paper needs to discuss early related works (missing the related works section)."
>
> If earlier works worked on learnable delays in the broader RNN community, we do agree that they should be cited. After reading this comment we found some works that could be cited in this paper:
> [2] “Adaptive time-delay neural network for temporal correlation and prediction.” Daw-Tung Lin, Judith E. Dayhoff, Panos A. Ligomenides. Proceedings of SPIE: Applications of Artificial Neural Networks, 1992.
> [3]“Trajectory production with the adaptive time-delay neural network.” Daw-Tung Lin. Neural Networks, vol. 8, no. 3, pp. 447–461, 1995.
> [4] Marom, Emanuel et al. “Efficient Training of Recurrent Neural Network with Time Delays.” Neural networks : the official journal of the International Neural Network Society vol. 10,1 (1997): 51-59. doi:10.1016/s0893-6080(96)00072-x
> It would be of great help if the reviewer could provide any reference that we missed on this topic.
>
> 6) "Furthermore, the authors claim in the abstract that their work "paves the way for neuromorphic hardware deployment," but the entire paper provides no discussion or evidence related to hardware, which overstates the contribution."
>
> We think our work paves the way for neuromorphic hardware deployment, in the sense that existing neuromorphic chips could already implement our DelRec-trained networks for inference, because most of them can implement LIF neurons, recurrent connections, and delays, and that is all we need. See for example [5] for a recent implementation of recurrent SNNs with delays on Loihi 2.
>
> [5] Mészáros, B., Knight, J. C., Timcheck, J., & Nowotny, T. (2025). A Complete Pipeline for deploying SNNs with Synaptic Delays on Loihi 2. arXiv preprint arXiv:2510.13757. https://arxiv.org/abs/2510.13757
>
> 7) "There are multiple instances of unclear symbols and formula definitions in the paper. For example, $\text{supp}(h_{\sigma, d})$ in Equation (12) and Rec. Delays in Table 1 and 2 are undefined in the main text, seriously affecting the readability and reproducibility of the paper."
>
> The symbol $\text{supp}(h_{\sigma, d}(\tau))$ is introduced with the sentence: ' the function $h_{\sigma, d}(\tau)$ has a finite support $\text{supp}(h_{\sigma, d}(\tau))$'. Therefore, $\text{supp}(h_{\sigma, d}(\tau))$ is the support of $h_{\sigma, d}(\tau)$. Moreover, we indeed did not specify that 'Rec. Delays' meant 'Recurrent delays', as we believed that is was clear enough knowing the topic of the article. We would be glad to provide a table with clearer definitions of symbols and abbreviations if the reviewer lets us know of other misunderstood elements. Yet, we believe that the claim: 'seriously affecting the readability and reproducibility of the paper' is seriously overstated.

---

> > ### Comment · Reviewer_izGg · 2025-11-19
> >
> > Thank you for your detailed response. You have addressed most of my major concerns, and I have raised my score. I strongly recommend that you integrate these key clarifications into the final revised manuscript. Furthermore, to prevent similar misunderstandings, I also suggest that you explicitly list your contributions in the introduction.

---

> ### Author Response · Authors · 2025-11-25
>
> Dear reviewer, we sincerely thank you for helping us improve our work by addressing the points you raised. These points will be added to our work. We also thank you for the consideration of our feedback, and your suggestion for our introduction. Please let us know if we can address any other concern you might have about our work.

---

### Note · Authors · 2025-12-11

I have read and agree with the venue's withdrawal policy on behalf of myself and my co-authors.